# 'No' Matters: Out-of-Distribution Detection in Multimodality Long Dialogue

## Abstract

Out-of-distribution (OOD) detection in multimodal contexts is essential for identifying deviations in combined inputs from different modalities, particularly in applications like open-domain dialogue systems or real-life dialogue interactions. This paper aims to improve the user experience that involves multi-round long dialogues by efficiently detecting OOD dialogues and images. We introduce a novel scoring framework named **D**ialogue **I**mage **A**ligning and **E**nhancing **F**ramework (DIAEF) that integrates the visual language models with the novel proposed scores that detect OOD in two key scenarios (1) mismatches between the dialogue and image input pair and (2) input pairs with previously unseen labels. Our experimental results, derived from various benchmarks, demonstrate that integrating image and multi-round dialogue OOD detection is more effective with previously unseen labels than using either modality independently. In the presence of mismatched pairs, our proposed score effectively identifies these mismatches and demonstrates strong robustness in long dialogues. This approach enhances domain-aware, adaptive conversational agents and establishes baselines for future studies.[1]

## 1 Introduction

In the regime of multimodal learning contexts, Out-Of-Distribution (OOD) detection involves identifying whether some unknown inputs from different modalities (e.g., text and images) deviate significantly from the patterns in the previously seen data. Specifically, an OOD instance under the multimodal setting is defined as one that does not conform to a certain distribution of interest, either by deviating in one modality or by showing the discrepancy across different modalities (Arora et al., 2021; Chen et al., 2021; Feng et al., 2022; Hsu et al., 2020). This is crucial in applications such as dialogue-image systems where the synergy between spoken or written language and visual elements is expected to adhere to certain semantic and contextual norms when identifying the In-Distribution (ID) pairs where they come from some known distribution.

Particularly in the dialogue system with inputs from different modalities, efficiently handling OOD queries/images can significantly improve user satisfaction and trust as the response quality hinges tightly on the understanding of the semantics from different modality inputs. Recognizing and managing OOD queries — those that deviate from expected dialogue or image patterns and contents — is essential for maintaining these systems' reliability and user experience, especially in dynamically changing dialogue systems with real-life interaction from users with much noise (Gao et al., 2024a;b). Taking three motivating examples as shown in Figure 1, we are given several dialogue-image pairs for OOD detection where our ID label is 'cat'. We will then consider two typical OOD cases in dialogue systems where either: 1) the dialogue label and image labels are not matched, or 2) even if the dialogue and image match, their labels might not be previously seen in the given data.

To effectively detect OOD samples in such a novel multi-modalities multi-round long dialogue scenario, we introduce **D**ialogue **I**mage **A**ligning and **E**nhancing **F**ramework (DIAEF), a framework that incorporates a novel OOD score for taking the first attempt on dialogue-image OOD detection for long dialogue systems. We propose a new score design across these two modalities, enabling more targeted controls for misalignment detection and performance enhancement.

---

[1] Code can be found in https://anonymous.4open.science/r/multimodal_ood-E443/README.md.

Such a framework could effectively boost anomaly detection and give better response strategies in long dialogue systems with interactive aims. This comprehensive score framework not only advances the field of multimodal conversations but also sets a new standard for domain-aware, adaptive long dialogue agent building for the future. To show the effectiveness of the proposed framework, we constructed a dataset consisting of over 120K dialogues in the multi-round application Question-answering systems and open-domain real interactive dialogues (Seo et al., 2017; Lee et al., 2021). Leveraging these dialogue datasets, we apply our proposed framework and demonstrate the effectiveness of the novel score design through various experiments. These experiments establish fundamental benchmarks and pave the way for future explorations in such a novel dialogue setting. Further-

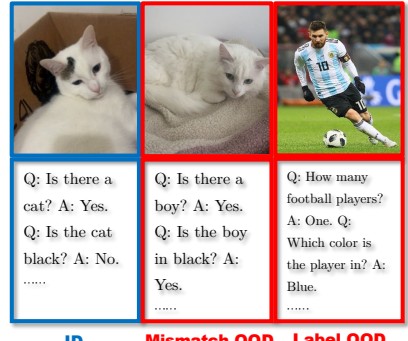

Figure 1: Motivating examples for ID, mismatched OOD and label OOD pair where the ID label is 'cat' and OOD label is 'sport'.

more, we integrate the crucial aspect of OOD detection, emphasizing its significance for enhancing the robustness and applicability of multimodal dialogue systems (Dai et al., 2023; Dosyn et al., 2022; Wu et al., 2024). To summarize, **our contributions** are listed as follows:

• We take the first attempt for OOD detection with the dialogues and propose a novel framework that enhances the OOD detection in cross-modal contexts, particularly focusing on scenarios where dialogue-image pairs either do not match with the semantics or even match, but their semantic labels are outside the known set, which matters in long-dialogue context for users.

• Our framework incorporates a novel scoring method by combining both dialogues and images to enhance the OOD detection while recognizing the mismatch pairs with the dialogue-image similarity.

• We demonstrate the practical application of our methods with the real-world multi-round long dialogue dataset, showcasing improvements in user experience and system reliability upon response. Further, our work establishes foundational benchmarks and methodologies that can serve as baseline standards for future research in the field of cross-modal detection on interactive dialogue systems.

## 2 RELATED WORK

**OOD Detection in Dialogue Systems.** Dialogue systems have become fundamental in applications ranging from virtual assistants and customer service bots to educational platforms with continuous multi-rounds (Feng et al., 2022; Kottur et al., 2019; Seo et al., 2017; Yu et al., 2019; Gao & Wang, 2024). The evolution of dialogue systems has seen a progression from rule-based and template-based approaches to statistical and machine learning methods (Zheng et al., 2020; Lang et al., 2023; Deka et al., 2023; Mei et al., 2024; Arora et al., 2021). Modern systems, particularly those based on deep learning models like BERT and GPT, have set new performance benchmarks (Yuan et al., 2024; Hendrycks et al., 2020; Yang et al., 2022; Ye et al., 2023). However, the complexity of these systems introduces challenges in understanding context and handling ambiguous semantic queries, necessitating more sophisticated approaches to maintain dialogue coherence and accuracy in interactive dialogue contexts, especially for real-life cases. OOD detection is a critical aspect of dialogue systems, ensuring their robustness and reliability in generating responses to user queries (Niu & Zhang, 2021; Chen et al., 2022). When dialogue systems encounter inputs that deviate from the training data distribution in long multi-round data, they risk generating incorrect or nonsensical answers, leading to user frustration and decreased trust. Effective OOD detection helps identify such anomalous queries, allowing the system to gracefully handle or reject them, thereby maintaining the quality and consistency of responses (Li & Lin, 2021), including softmax probability thresholding (Liu et al., 2023; Dhuliawala et al., 2023), auxiliary models (Wang et al., 2024; Zheng et al., 2024; Ramé et al., 2023), generative models (Cai & Li, 2023; Ktena et al., 2024; Graham et al., 2023), and self-supervised learning (Azizi et al., 2023; Wallin et al., 2024; Liu et al., 2021). The integration of effective OOD detection mechanisms is crucial for the continued advancement and trustworthiness of QA dialogue systems (Salvador et al., 2017; Feng et al., 2022).

**Dialogue-based Multimodality OOD Detection.** Due to the complexity of dialogue in multi-turns and information complexity embedded in the connection of preceding turns within the long dialogue, successfully detecting whether the information from the dialogue and images are within the same domain stands as a technical challenge in OOD detection (Fort et al., 2021; Basart et al., 2022). Previous works attempted to evaluate the generated pseudo-OOD samples' impact on the OOD section in dialogue settings (Marciniak, 2020), which improved OOD detection performance after introducing the generated dialogues when utilizing unlabeled data, making the model practical and effective for real-world applications (Zheng et al., 2020). Later studies used the information bottleneck principle to extract robust representations by filtering out irrelevant information for multi-turn dialogue contexts (Lang et al., 2023). Furthermore, the crucial aspect of OOD detection in multimodal long dialogue is still under investigation, emphasizing the significance of multimodal conversational user experience in question-answering systems.

**Multi-label OOD Detection.** While numerous studies have improved approaches for multi-class OOD detection tasks, investigating multi-label OOD detection tasks has been notably limited. A recent advancement is the introduction of Spectral Normalized Joint Energy (SNoJoE) (Mei et al., 2024), a method that consolidates label-specific information across multiple labels using an energy-based function. Later on, the sparse label co-occurrence scoring (SLCS) leverages these properties by filtering low-confidence logits to enhance label sparsity and weighting preserved logits by label co-occurrence information (Wang et al., 2022). Considering the vision-language information as input in models like CLIP (Radford et al., 2021), traditional vision-language prompt learning methods face limitations due to ID-irrelevant information in text embeddings. To address this, the Local regularized Context Optimization (LoCoOp) approach enhances OOD detection by leveraging CLIP's local features in one-shot settings (Miyai et al., 2024). However, previous approaches majorly implied the limitation only in computer vision tasks without focus on dialogue or Natural Language Processing tasks(Wei et al., 2015; Zhang & Taneva-Popova, 2023; Wang et al., 2021; 2022).

## 3 PROBLEM FORMULATION

To formally define the cross-modal OOD problem, we focus on the detection with dialogue and image pairs within a multi-class classification framework. Specifically, we have a batch of $N$ pairs of images and dialogues, along with their labels, denoted by $\{(i_n, t_n), \mathbf{y}_n\}_{n=1}^N$ where $i_n \in \mathcal{I}$ and $t_n \in \mathcal{T}$ denote the input image and dialogues and $\mathcal{I}$ and $\mathcal{T}$ are the image and dialogue spaces, respectively. Here, the instance pair may be associated with multiple labels $\mathbf{y}_n$ with $\mathbf{y}_n = \{y_{n,1}, y_{n,2}, \cdots, y_{n,K}\} \in [0,1]^K$ where $y_{n,k} = 1$ if the dialogue-image pair is associated with $k$-th label and $K$ denotes the total number of in-domain categories. Our proposed score function enhances the ability to distinguish between ID and OOD data cross-joint detection for image and dialogue, making it applicable in multimodality scenarios. Based on this setup, the goal of the OOD detection is to define a decision function $G$ such that:

$$G(i, t, \mathbf{y}) = \begin{cases} 0 & \text{if } (i, t, \mathbf{y}) \sim \mathcal{D}_{\text{out}}, \\ 1 & \text{if } (i, t, \mathbf{y}) \sim \mathcal{D}_{\text{in}}. \end{cases} \tag{1}$$

**Remark 1** *Different from unimodal OOD detection (Lee et al., 2018; Basart et al., 2022; Hendrycks & Gimpel, 2016; Du et al., 2022; Wu et al., 2023), in the cross-modal detection scenarios, we need to additionally consider whether the image and dialogue come from the same distribution, i.e., whether the image and dialogue are semantically matched in the interaction context. In particular, we will consider several scenarios for detecting OOD samples: 1). the image and dialogue do not match (e.g., in terms of content or description), and 2). the in-domain sample does not contain any out-of-domain labels, meaning previously unseen labels appear, or 3). both cases occur simultaneously.*

To determine $G$ in practice, we may need to consider the relationship between dialogue and images additionally. To this end, let $M : \mathcal{I} \cup \mathcal{T} \to \mathbb{R}^d$ be a vision-language model that could encode the image $i_n$ with the image embedding $x_{i,n} \in \mathbb{R}^d$, and the dialogues with the text embedding $x_{t,n} \in \mathbb{R}^d$ in the same latent space as in the image. To classify the relevance of an image to a dialogue according to the label $\mathbf{y}_n$, we first use a scoring function $s : \mathbb{R}^d \times \mathbb{R}^d \to \mathbb{R}$, which evaluates the similarity or relevance between the image and text embeddings from $M$. We then further compare these two embeddings with the label $\mathbf{y}_n$ with the dialogue score function $s_T : \mathbb{R}^d \times [0,1]^K \to \mathbb{R}$ and image score function $s_I : \mathbb{R}^d \times [0,1]^K \to \mathbb{R}$. For simplicity, we use $s_I$ (or $s_T$) interchangeably with

$s_I(x, \mathbf{y})$ throughout the paper. Finally, we could conduct a fusion on the three scores $g(s, s_T, s_I)$ for some fusion function $g$ and check if the numeric value exceeds $\lambda$ to determine whether it is in-domain or out-of-domain. Given the above definition, given a dialogue-image data pair $(i, t)$, we will examine whether it is ID or OOD per dialogue-image pair in the given label set $\mathcal{Y}$ with the following criterion.

**Definition 1 (Cross-Modal OOD Detection)** *We use the following detection criterion for out-of-domain samples.*

- ***In-domain:*** *given both embeddings $x_i$ from the images and $x_t$ from the dialogue, and a certain label $y$. We say the image is in-domain with the dialogue if $g(s(x_i, x_t), s_t(x_t, \mathbf{y}), s_i(x_i, \mathbf{y})) \geq \lambda$.*

- ***Out-of-domain::*** *given both embeddings $x_i$ from the images and $x_t$ from the dialogue, we say the image is out-of-domain with the dialogue if $g(s(x_i, x_t), s_t(x_t, \mathbf{y}), s_i(x_i, \mathbf{y})) < \lambda$.*

*for some fusion function $g$ and some threshold $\lambda$.*

## 4 DIALOGUE IMAGE ALIGNING AND ENHANCING FRAMEWORK

To intuitively demonstrate our framework, we draw the overall workflow in Figure 2. The workflow consists of three parts: in the first stage, we will employ a vision language model, such as CLIP (Radford et al., 2021) and BLIP (Li et al., 2022), to derive meaningful descriptors or feature embeddings from images and dialogues, respectively. Note that the model we used here would map the image and dialogue into the same latent space so that the similarity between the two can be easily calculated. These processes yield embeddings $x_I$ for images and $x_T$ for dialogues. Then, utilizing these embeddings, we apply a scoring function $s(x_I, x_T)$ to assess the relevance between an image and a dialogue. The outcome of this function helps us determine whether the dialogue-image pair falls within the categories, indicating a high relevance in semantics, or the out-of-distribution categories with mismatches, suggesting low or no relevance.

In addition to this score, we will further train two label extractors to compare the whole pair with the label set to determine if the pair is in-distribution or out-of-distribution using $s_I(x_I, \mathbf{y})$ that evaluates the similarity between the image and the label and $s_T(x_T, \mathbf{y})$ that evaluates the similarity between the text and the label. We will use conventional methods to combine these two scores and determine whether the pair is ID or OOD based on the threshold $\lambda$.

This paper aims to enhance the detection of OOD samples by combining dialogues and images and identifying the misalignments between them. To this end, we naturally propose the DIAEF score function in general:

$$g(x_T, x_I, \mathbf{y}; s, s_T, s_I) = s(x_T, x_I)^\gamma (\alpha s_I(x_I, \mathbf{y}) + (1 - \alpha) s_T(x_T, \mathbf{y})), \tag{2}$$

where the first part $s(x_T, x_I)^\gamma$, which we call the alignment term, controls the similarity between the image and the dialogue. If the image and dialogue are highly similar, this term will be large and vice versa. This allows us to identify the misalignment between images and dialogues in a long dialogue system. The second part $(\alpha s_I + (1 - \alpha) s_T)$, namely the enhancing term, enhances the detection of OOD samples by linearly combining the dialogue and image scores, where the weighting hyperparameter $\alpha$

Table 1: OOD Scores for $s_I/s_T$.

| Method | Score |
|---|---|
| Probability | $P_y(x)$ |
| MSP (Hendrycks & Gimpel, 2016) | $\max_{y \in \mathcal{Y}} \frac{f_y(x)}{\sum_y f_y(x)}$ |
| Logits (Hendrycks et al., 2019) | $f_y(x)$ |
| Energy (Wang et al., 2021) | $\log(1 + e^{f_y(x)})$ |
| ODIN (Liang et al., 2017) | $f_y(x + \epsilon \Delta)/T$ |
| Mahalanobis (Lee et al., 2018) | $(x - \mu_y)^T \Sigma_y^{-1} (x - \mu_y)$ |

controls the relative importance of the image: if $\alpha$ is selected to be large, we rely more on images for OOD detection; conversely for a small $\alpha$, we rely more on the dialogue. The purpose of using a multiplicative combination of the alignment and enhancing terms is: (1) identifying mismatched OOD pairs where either the image or dialogue might have high relevance to the label, making the enhancing term potentially large (depending on $s_I$ or $s_T$). To identify these pairs as OOD samples, we naturally multiply the enhancing term by $s(x_T, x_I)$; (2) identifying matched pairs with OOD labels where $s(x_T, x_I)$ may be large, but the enhancement term is likely to be small since the image

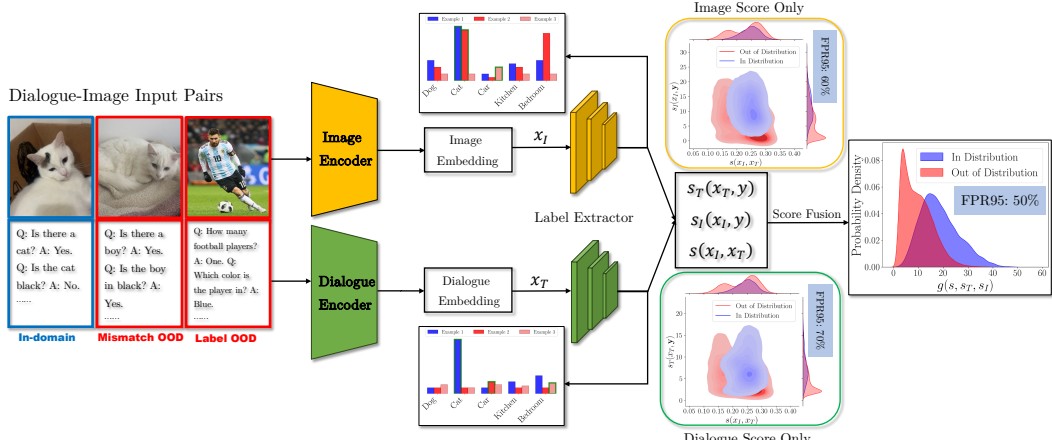

Figure 2: The workflow for three motivating examples for cross-modal OOD detection, including ID pair, mismatched OOD pair, and label OOD pair. The workflow consists of three main parts: the dialogue and image will be firstly processed and passed into a visual language model to get the image and dialogue embeddings; then two label extractors will be trained on both the image and dialogue embeddings for predictions and score calculations; finally the score function $s$, $s_T$ and $s_I$ are aggregated to determine the threshold $\lambda$ at recall rate of 95%. The FPR95% is reported to demonstrate that combining images and dialogue outperforms using images or dialogue alone.

and dialogue have low relevance to the label. To show this mathematically, we give a theoretical justification for the proposed score in Appendix B.

The choice of the functions $s(x_I, x_T)$ depends on the selection of the trained visual language model. For example, in CLIP, contrastive loss is used to measure the similarity between images and text (dialogue) based on cosine similarity (Radford et al., 2021). Similarly, BLIP employs image-text matching loss and leverages cosine similarity to align the representations of images and text (Li et al., 2022). With those two models, selecting cosine similarity as an appropriate score for $s(x_I, x_T)$ is natural. Regarding $s_I$ and $s_T$, which measure the scores between embeddings and labels, various potential choices and aggregation methods are available. For example, one direct way is to use the probability of the model output $P_y(x)$ as the score for the category $y$ with the input $x$, and we could further aggregate the probability over all categories using sum or max methods to derive our final DIEAF score. More complicated scores would involve some probability transformation, such as the logits $f_y(x)$ used in (Hendrycks et al., 2019) or the normalized version called MSP as used in (Hendrycks & Gimpel, 2016). Some other effective scores would involve the pre-trained models, such as the ODIN method proposed in (Liang et al., 2017) modifies the input by adding a gradient-based perturbation, or the method proposed in (Lee et al., 2018) computes the Mahalanobis distance between the embeddings from the pre-trained model and the class conditional distributions in the feature space. Table 1 shows the list of possible scores that could fit in our framework.

## 5 EXPERIMENTS

In this section, we evaluate DIAEF and other baselines using several datasets.

### 5.1 EXPERIMENTAL SETUP

**Datasets.** In this section, we utilize the Visdial dataset (Das et al., 2017) and Real MMD dataset (Lee et al., 2021) for OOD detection in long dialogue systems. The Visdial dataset comprises over 120K images sourced from the COCO image dataset (Lin et al., 2014), coupled with collected multi-round dialogues in a one-to-one mapping format between modalities. We constructed a testing multi-round question-answering dataset with full semantic context to evaluate our OOD detection methods, including all dialogue-image pairs and an additional 10K mismatched pairs. Each entry in this dataset contains an image, a full conversation, and a set of labels with 80 specific categories.

The dataset is further organized into 12 higher-level supercategories: *animal*, *person*, *kitchen*, *food*, *sports*, *electronic*, *accessory*, *furniture*, *indoor*, *appliance*, *vehicle*, and *outdoor*. Another related dataset, called the Real MMD dataset, contains images sourced from COCO (Lin et al., 2014) and texts from different sources such as DailyDialog (Li et al., 2017) and Persona-Chat (Miller et al., 2017), meaning they may not be perfectly matched but instead have a certain degree of similarity. The dataset statistics are presented in Table 7 and Table 8 in Appendix A.

**OOD Label Selection.** In our study, we propose a label selection score function for selecting OOD labels that effectively combines semantic distance (Huang et al., 2008; Kadhim et al., 2014; Li & Han, 2013; Rahutomo et al., 2012; Lahitani et al., 2016) and ontological hierarchy via the WordNet path calculation (Aminu et al., 2021; Dosyn et al., 2022; Fellbaum, 2010; Marciniak, 2020; Martin, 1995). The score function integrates multiple criteria to enhance the robustness and accuracy of OOD detection. Semantic distance is quantified using cosine similarity between vector representations of candidate labels and the remaining labels in the label set. We compute the maximum cosine similarity to any ID label for each candidate OOD label and select those with values below a predefined threshold, ensuring semantic distinctiveness. Additionally, we leverage ontological hierarchies, such as WordNet, to measure the path length between candidate labels and ID labels. Candidates with a minimum path length exceeding a specified threshold are selected, ensuring they are not closely related in the hierarchy. This dual-criteria approach ensures that selected OOD labels are both semantically distant and ontologically distinct from ID labels, enhancing the efficacy of the OOD detection system. By integrating these methods, our score function effectively mimics real scenarios where the OOD labels generally differ from the ID labels[2]. Therefore, we define the selection score as:

Table 2: Top 5 Labels

| Label | $S(c)$ |
|---|---|
| Animal | 5.12 |
| Person | 5.01 |
| Sports | 4.89 |
| Vehicle | 4.80 |
| Outdoor | 4.79 |

$$S(c) = w_1 \sum_{y \in \mathcal{Y} \setminus c} (1 - S_{\cos}(M(c), M(y))) + w_2 \sum_{y \in \mathcal{Y} \setminus c} (1 - S_{\text{PATH}}(c, y)), \qquad (3)$$

where

$$S_{\cos}(\mathbf{a}, \mathbf{b}) = \frac{\mathbf{a} \cdot \mathbf{b}}{\|\mathbf{a}\|\|\mathbf{b}\|}, \quad S_{\text{PATH}}(y_1, y_2) = \frac{1}{1 + \ell_d(y_1, y_2)}. \qquad (4)$$

Here, $M(c)$ and $M(y)$ are the vector representations of the candidate OOD labels $c$ and the ID label $y$ with the encoder $M$, respectively, $w_1$ and $w_2$ are the weights assigned to each criterion, and $\mathcal{Y}$ represents the total valid label set. The term $S_{\cos}$ measures the semantic distance, and $S_{\text{PATH}}(y_1, y_2)$ measures the ontological distance between labels with the path distance $\ell_d(y_1, y_2)$ between the words $y_1$ and $y_2$ in the WordNet. We conduct the score selection on the Visdial dataset with $w_1 = w_2 = 0.5$, and the top five scores with the most distinguished labels are shown in Table 2. To ensure that the selected OOD labels are both semantically distant and ontologically distinct from ID labels, we select candidates $c$ where the score $S(c)$ is the highest.

**Experiment Details.** Based on Table 2, we select the label '*animal*' as the OOD label to show the framework's effectiveness. We will have 95K ID pairs and 37K OOD pairs for QA dataset and 12.7K ID pairs and 12.2K OOD pairs for the Real MMD dataset. We will use the 8:2 train-test split, yielding 77K/54K and 10.2K/14.7K train/test pairs, respectively. For encoders for images and dialogues, we use CLIP ViT-B/32 (Radford et al., 2021) throughout the experiments, and we trained the label extractors with the ID training sample with a 5-layer fully connected network. More details are given in Appendix A. Additionally, we use sum and max aggregation methods for the above methods. The sum aggregation method combines the scores across all considered classes or components, providing an overall score that reflects the cumulative effect. The max aggregation method selects the maximum

---

[2]For tuning label selections, we list the table below using the cosine similarity (Descending order): [sports, outdoor, animal, fashion, electronics, person, bedroom, vehicle, appliance, kitchen, food, furniture]. With wordnet only: [person, animal, vehicle, furniture, appliance, kitchen, food, bedroom, fashion, electricity, outdoor, sports]. Using only cosine similarity, labels skewed towards broad, abstract categories like "sports" and "outdoor", reflecting a focus on general semantic similarities (complex context where more background information is needed). Comparably, using only WordNet similarity emphasized specific, taxonomically grounded categories like "person" and "animal", highlighting hierarchical relationships (suitable when labels are short descriptors). Adaptive weighting or context-specific tuning could be explored for future refinements where weights are dynamically adjusted regarding dataset characteristics or task requirements.

Table 3: The comparison of OOD detection performance with QA dataset under CLIP extraction and different scores. **Bold** numbers are superior results for each DIAEF score and aggregation method. Metrics reported in % include FPR95 (↓ indicates the lower the better), AUROC, and AUPR (↑ indicates the higher the better).

| | | FPR95↓ / AUROC↑ / AUPR↑ | | |
| --- | --- | --- | --- | --- |
| OOD Scores | Aggregation | Baseline w/ OOD Scores | | DIAEF w/ OOD Scores |
| | | Image | Dialogue | |
| MSP | Max | 84.4/ 64.8/ 49.0 | 76.9/ 66.5/ 48.8 | **73.4/ 73.2/ 53.5** |
| Prob | Max | 60.0 / 75.6 / **57.9** | 67.9 / 73.5 / 56.1 | **55.3 / 78.8 / 57.9** |
| | Sum | **70.7** / 68.3 / 49.0 | 91.9 / 62.3 / 45.7 | 72.8 / **73.6 / 56.6** |
| Logits | Max | 60.0 / 75.6 / 57.9 | 67.9 / 73.5 / 56.1 | **57.2 / 82.6 / 72.7** |
| | Sum | **91.2 / 59.2 / 43.6** | 98.6 / 44.1 / 36.0 | 97.2 / 49.9 / 37.4 |
| ODIN | Max | **59.1** / 75.4 / 57.6 | 72.1 / 73.2 / 55.5 | 59.6 / **78.9 / 58.8** |
| | Sum | **71.2** / 68.0 / 48.8 | 91.9 / 61.6 / 45.2 | 73.0 / **73.2 / 56.0** |
| Mahalanobis | Max | **49.2** / 81.3 / 62.9 | 66.0 / 75.8 / 56.8 | 49.7 / **83.2 / 67.1** |
| | Sum | 88.5 / 75.5 / 57.5 | 78.6 / 68.6 / 50.0 | **75.0 / 76.2 / 60.2** |
| JointEnergy | Max | 60.0 / 75.6 / 57.9 | 67.9 / 73.5 / 56.1 | **57.6 / 82.5 / 72.6** |
| | Sum | 58.3 / 75.8 / 58.0 | 67.0 / 74.1 / 57.1 | **55.9 / 82.3 / 72.2** |
| Average | Max | 62.1 / 74.7 / 57.2 | 69.8 / 72.7 / 54.9 | **58.8 / 79.9 / 63.8** |
| | Sum | 76.0 / 69.4 / 51.4 | 85.6 / 62.1 / 46.8 | **74.8 / 71.0 / 56.5** |

score among all classes or components, highlighting the strongest single match. These aggregation methods allow us to assess the performance and robustness of each scoring function comprehensively. We use the cosine similarity for $s(x_I, x_T)$ for CLIP embeddings and set $\gamma = 1$ and $\alpha = 0.5$ as the hyperparameter default values. To ensure the consistency and reliability of our results, all experiments were executed on a system featuring a single NVIDIA RTX 2080 Super GPU.

**Adopted OOD Scores.** Throughout the experiments, we used several general OOD scores to evaluate the effectiveness of the framework, which includes Probability (Prob), Maximum Softmax Probability (MSP) (Hendrycks & Gimpel, 2016), Logits (Hendrycks et al., 2019), Joint Energy (Wang et al., 2021), ODIN (Liang et al., 2017) and Mahalanobis distance (Lee et al., 2018). These baseline methods provide a diverse set of techniques for OOD detection, each leveraging different aspects of the model's output and feature embeddings. Then, we included two baselines with the DIEAF scores in our evaluation. The first baseline, with image only, utilizes the score function $s_I(x_I, \mathbf{y})$ to determine the score threshold for OOD. The second baseline, with dialogue only, employs a similar approach, using the score function $s_T(x_T, \mathbf{y})$ instead. All methods are evaluated with the metrics FPR95, AUROC and AUPR as previously mentioned in Section 4.

**Evaluation.** We include the following metrics in our evaluation for OOD detection: FPR95, AUROC and AUPR. FPR95 measures the rate at which false positives occur when the true positive rate is fixed at 95%. This metric indicates how often the model incorrectly classifies a negative instance as positive when it correctly identifies 95% of all positive instances; a lower FPR95 value signifies a better-performing model. AUROC evaluates the overall ability of a model to discriminate between positive and negative classes across all possible classification thresholds. It involves plotting the ROC curve with the true positive rate against the false positive rate at various threshold settings. A higher AUROC value denotes a better-performing model. AUPR, similar to AUROC, focuses on the precision-recall curve, which plots precision against recall. This metric is particularly useful in class imbalance scenarios. A better AUPR indicates a better model's performance.

## 5.2 MAIN RESULTS

With the aforementioned experimental settings, we evaluate various DIAEF scores on the given QA and Real MMD datasets and report the performance results in Table 3 and 4. The tables show that our proposed framework generally outperforms the results obtained using only images or dialogue across most metrics. In particular, the joint energy and Mahalanobis scores with the sum or max aggregation consistently perform well across most metrics. In addition, the naive probability and ODIN scores also show competitive performance. Interestingly, the max aggregation method tends to be more

Table 4: The comparison of OOD detection performance with Real MMD dataset under CLIP extraction and different scores.

| OOD Scores | Aggregation | FPR95↓ / AUROC↑ / AUPR↑ | | DIAEF |
|---|---|---|---|---|
| | | Baseline w/ OOD Scores | | w/ OOD Scores |
| | | Image | Dialogue | |
| MSP | Max | 91.1/56.2/19.4 | 94.5/52.5/18.9 | **85.8/69.2/32.9** |
| Prob | Max | 79.2/64.1/22.9 | 93.4/53.7/19.3 | **75.8/74.0/36.0** |
| | Sum | 90.6/58.2/21.1 | 94.4/51.9/18.7 | **83.0/69.7/33.1** |
| Logits | Max | **79.2**/64.1/22.9 | 93.4/53.7/19.3 | 84.8/**70.9/34.1** |
| | Sum | **94.5/49.0/17.8** | 97.3/47.6/17.2 | 98.8/38.6/14.3 |
| ODIN | Max | 79.6/64.3/23.4 | 94.0/53.4/19.3 | **75.3/74.4/36.9** |
| | Sum | 91.1/57.1/20.8 | 94.9/51.3/18.5 | **82.2/69.2/32.0** |
| Mahalanobis | Max | **54.9**/69.9/26.1 | 93.5/51.2/17.6 | 63.5/**76.8/36.2** |
| | Sum | 93.3/66.0/25.2 | 94.2/49.1/16.9 | **86.6/73.3/36.5** |
| JointEnergy | Max | **79.2**/64.1/22.9 | 93.4/53.7/19.3 | 83.5/**71.6/34.3** |
| | Sum | **79.5**/64.9/24.4 | 93.6/54.2/19.5 | 80.5/**72.8/37.4** |
| Average | Max | **77.2**/63.8/22.9 | 93.7/53.0/19.0 | 78.1/**72.8/35.1** |
| | Sum | 89.8/59.0/21.9 | 94.9/50.8/18.2 | **86.2/64.7/30.7** |

effective than the sum method. This could be because we are dealing with a multi-label problem. Adding up scores for all labels might introduce more noise, which confuses the OOD and ID scores and thus reduces detection performance. For dialogues, the performance is not as good as for images. This is because dialogues contain some noises, such as stopwords, that are unrelated to the labels, whereas images with segmentation are more directly related to the labels. However, even though the dialogue alone may not perform well, combining it with images could significantly enhance the OOD detection performance. The results demonstrate that DIEAF performs effectively when combining dialogue and image scores, especially when introducing mismatched pairs.

## 5.3 ANALYSIS OF EXPERIMENTAL RESULTS

To gain deeper insights into the proposed framework, we conduct several ablation studies to examine the impact of mismatched pairs, the effectiveness of $s(x_I, x_T)$, and the choices of $\alpha$ and $\gamma$.

**Effect of Mismatched Pairs.** To investigate the effect of the mismatched pairs, we conduct the experiments with the same setting by excluding the mismatched pair in the testing set and report the results in Table 5. Here, we only report FPR95 for simplicity and also compare the results by setting $\gamma = 0$ without introducing the dialogue-image similarity.

From the table, it can be observed that when there are no mismatched pairs, setting $\gamma$ to 1 can actually harm our results to some extent. This is because, for OOD pairs without mismatched pairs, their similarity score $s(x_I, x_T)$ can still be high. In such cases, multiplying by the similarity can adversely affect OOD results. Setting $\gamma$ to 0 in these situations improves FPR95 results for most cases, indicating that simply combining image and dialogue modalities, even without mismatched pairs, performs better than the unimodality. Additionally, comparing Table 3 and 5, we see that introducing mismatched pairs generally leads to worse performance than having no mismatched pairs. This demonstrates that mismatched pairs indeed pose a challenge for OOD detection. To achieve better results, we will further study the impact of $\gamma$ and $\alpha$ to optimize OOD detection performance.

**Effect of VLM models.** We further tested the performance of the DIAEF score function with the BLIP model (Li et al., 2022) under the same setting as CLIP (also see details in Appendix A), and we report the results in Table 6. Even with BLIP, the pattern is still maintained as the proposed score achieves better performance compared to the single modality, and the framework handles mismatched and previously unseen OOD scenarios.

**Effect of $s(x_I, x_T)$.** We draw Figure 5 for image scores as an illustration that consists of three subplots showing the change of score distribution with $s(x_I, x_T)$ introduced. Here Figures 5a and 5c present the distribution of $s_I(x_T, x)$ and $s_I(x_T, x)s(x_I, X_T)$ for both ID and OOD data with FPR95 highlighted, respectively. Figure 5b displays the joint distribution of $P(s, s_I)$ for both ID and OOD

Table 5: The comparison of **FPR95** performance (the lower the better) in % with DIEAF framework under different scores **without** any mismatched pairs. **Bold** numbers are superior results for each DIAEF score and aggregation method.

| OOD Scores | Aggregation | Baseline | | DIAEF ($\gamma = 0$) | DIAEF ($\gamma = 1$) |
| --- | --- | --- | --- | --- | --- |
| | | Image | Dialogue | | |
| MSP | Max | 81.2 | 71.4 | **69.4** | 81.4 |
| Prob | Max | 49.7 | 59.9 | **46.6** | 64.4 |
| | Sum | **63.6** | 91.2 | 72.7 | 77.1 |
| Logits | Max | 49.7 | 59.9 | **45.7** | 47.6 |
| | Sum | **90.1** | 99.7 | 98.1 | 96.2 |
| ODIN | Max | **48.5** | 65.4 | **48.5** | 69.2 |
| | Sum | **64.2** | 91.2 | 72.4 | 79.3 |
| Mahalanobis | Max | 35.5 | 57.5 | **34.3** | 37.8 |
| | Sum | 86.4 | 73.7 | 68.1 | **65.0** |
| JointEnergy | Max | 49.7 | 59.9 | **46.7** | 48.7 |
| | Sum | 47.4 | 58.6 | **45.7** | 47.6 |
| Average | Max | 52.4 | 62.3 | **48.5** | 58.1 |
| | Sum | **70.3** | 82.9 | 71.4 | 73.0 |

Table 6: The comparison of OOD detection performance with QA dataset under BLIP extraction and different scores.

| | | **FPR95↓ / AUROC↑ / AUPR↑** | | |
| --- | --- | --- | --- | --- |
| OOD Scores | Aggregation | Baseline | | DIAEF ($\gamma = 1$) |
| | | Image | Dialogue | |
| MSP | Max | 85.8/58.7/37.4 | 83.5/64.8/39.8 | **75.9/75.1/52.7** |
| Prob | Max | **64.3**/71.2/45.1 | 80.5/67.1/42.2 | 67.0/**78.7/56.5** |
| | Sum | 78.8/64.4/39.3 | 96.8/55.9/35.9 | **74.2/72.7/51.2** |
| Logits | Max | 64.3/71.2/45.1 | 80.5/67.1/42.2 | **62.9/80.9/63.8** |
| | Sum | **95.8/52.9/33.8** | 98.1/41.9/29.3 | 99.1/40.1/26.5 |
| ODIN | Max | **63.9**/71.1/44.9 | 81.4/67.2/42.1 | 67.7/**79.3/57.7** |
| | Sum | 79.1/64.2/39.2 | 97.0/56.1/36.0 | **74.5/72.5/50.9** |
| Mahalanobis | Max | **46.9**/77.7/50.6 | 81.0/66.9/40.5 | 52.6/**87.7/75.4** |
| | Sum | 79.7/71.5/46.2 | 92.5/59.0/35.9 | **67.6/78.7/61.0** |
| JointEnergy | Max | 64.3/71.2/45.1 | 80.5/67.1/42.2 | **62.8/81.0/63.8** |
| | Sum | 63.0/71.8/45.8 | 80.4/67.3/43.2 | **61.7/80.7/64.5** |
| Average | Max | 64.9/70.2/44.7 | 81.2/66.7/41.5 | **64.8/80.5/61.2** |
| | Sum | 79.3/65.0/40.9 | 93.0/56.0/36.1 | **75.4/68.9/50.8** |

data, with the x-axis representing the similarity score $s(x_I, x_T)$ and the y-axis representing the image score $s_I(x_I, \mathbf{y})$, with density indicated by colour intensity and marginal distributions shown as histograms. The figures show that without multiplying by $s(x_I, x_T)$, the distributions of ID and OOD are not well-separated, and the FPR95 is around 0.58. However, after applying the similarity score, the distributions of ID and OOD become more apart, and the FPR95 decreases to approximately 0.54. This occurs because, when examining the joint distribution, we find that for the ID data, most similarity values are around 0.25. In contrast, there are two peaks for the OOD data: one around 0.25 (for matched pairs) and another around 0.15 (for mismatched pairs). This indicates that if we multiply by this similarity, the mismatched OOD pairs would have lower scores, making distinguishing between ID and OOD easier.

**Effect of $\gamma$.** Intuitively, when $\gamma$ is smaller, similar and dissimilar dialogue-image pairs will have approximately the same alignment score. Conversely, when $\gamma$ is larger, the score differences between similar and dissimilar pairs become more pronounced, emphasizing the role of dialogue-image similarity in OOD detection. Therefore, we selected several values of $\gamma$ ranging from 0 (i.e., not using dialogue-image similarity) to 3 and plotted the curves under different score aggregation methods. Figure 3 shows that the optimal value of $\gamma$ varies significantly depending on the choice of score and

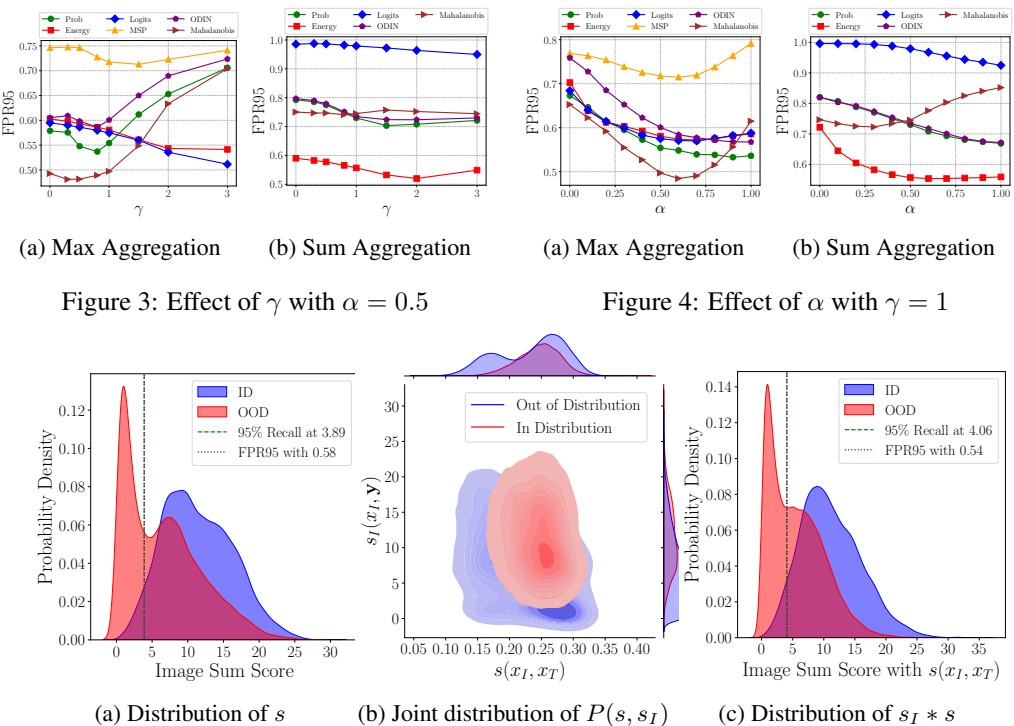

(a) Max Aggregation       (b) Sum Aggregation       (a) Max Aggregation       (b) Sum Aggregation

Figure 3: Effect of $\gamma$ with $\alpha = 0.5$       Figure 4: Effect of $\alpha$ with $\gamma = 1$

(a) Distribution of $s$       (b) Joint distribution of $P(s, s_I)$       (c) Distribution of $s_I * s$

Figure 5: An illustration of the effectiveness of $s(x_I, x_T)$

aggregation method. For instance, with max aggregation, most methods show a trend where the FPR95 initially decreases with increasing $\gamma$ and then rises again, with the optimal value around 1. However, the Energy and Logits methods show a trend of decreasing FPR95 as $\gamma$ increases, indicating these methods are more sensitive to misalignment. On the other hand, for the sum aggregation method, changing the $\gamma$ value has a limited effect on OOD detection. This could be because the sum method combines too much redundant label information, and the enhancement term plays a major role. If the enhancement term is not particularly effective, the impact of misalignment is minimal.

**Effect of $\alpha$.** When $\alpha$ is small, we place more emphasis on the image score along with the alignment term for OOD detection; conversely, when $\alpha$ is large, we emphasize more on the dialogue score. We plotted the results for different score aggregations in Figure 4. From the max aggregation results, we observe that using only the image or dialogue scores is not the most effective approach. Instead, combining both and selecting a value around 0.5 yields the best results, demonstrating the effectiveness of our framework. In the sum aggregation plot, we see that for most methods (except for Mahalanobis), the performance in terms of FPR95 improves as $\alpha$ increases. This indicates that images do not significantly contribute to recognition for the sum aggregation compared to dialogue.

## 6 CONCLUSION AND LIMITATION

This paper introduces a cross-modal OOD score framework, DIAEF, designed to expand OOD detection in cross-modal QA systems by integrating images and dialogues. DIAEF combines alignment scores between dialogue-image pairs with an enhancing term that leverages both the image and dialogue. Experimental results demonstrate DIAEF's superiority over baseline methods with general metrics such as FPR95 and show the framework's effectiveness. However, there are some spaces for future work. First, due to the scarcity of datasets, we initially validated our framework on VisDial and demonstrated its effectiveness. More dialogue-image datasets are worth exploring for validation. Second, the existing scores have proven the effectiveness of this framework, but further improvements could be achieved by applying some transformations or smoothing techniques to make the distributions of ID and OOD more distinct. Finally, this framework is applicable to more visual language models, such as multimodal models like BLIP, and can further enhance OOD performance using various image-text matching criteria.

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

## A EXPERIMENT DETAILS

The dataset stats are summarized as follows:

Table 7: Statistics of Visdial QA dataset

| Stats | Matched | Mismatched | ID | OOD |
|---|---|---|---|---|
| # Pair | 122K | 10K | 95K | 37K |
| # Train | 77K | 0 | 77K | 0 |
| # Test | 45K | 10K | 18K | 37K |
| # Turn per dialog | | 10 | | |
| # Categories | | 80 | | |
| # Supercategories | | 12 | | |

Table 8: Statistics of Real MMD dataset

| Stats | Matched | Mismatched | ID | OOD |
|---|---|---|---|---|
| # Pair | 17K | 8K | 12.7K | 12.2K |
| # Train | 10.2K | 0 | 10.2K | 0 |
| # Test | 14.7K | 8K | 2.5K | 12.2K |
| # Turn per dialog | | $5 \sim 15$ | | |
| # Categories | | 80 | | |
| # Supercategories | | 12 | | |

We give detailed experimental settings in the following table.

Table 9: Experimental Details

| Parameters | Configurations |
|---|---|
| $\gamma$ | 1 |
| $\alpha$ | 0.5 |
| Image Encoder | CLIP Vi-T B/32 or BLIP ITM Base |
| Dialogue Encoder | CLIP Vi-T B/32 or BLIP ITM Base |
| $s(x_I, x_T)$ | Cosine Similarity |
| Label Extractor | 5-Layer DNN with size [512/256, 256, 128, 64, 11] |
| Activation Function | Relu & Sigmoid |
| Batch Size | 32 |
| Learning Rate | 0.001 |
| Optimizer | Adam |
| ID label | *person, kitchen, food, sports, electronic, accessory, furniture, indoor, appliance, vehicle, outdoor* |
| OOD label | *animal* |
| $\eta$ in ODIN | 0.001 |
| $T$ in ODIN | 1 |
| Image Features in Mahalanobis | CLIP/BLIP(Image) |
| Text Features in Mahalanobis | CLIP/BLIP(Dialogue) |

## B THEORETICAL JUSTIFICATION

**Assumption 1** *We denote ID distribution as $P(x_I, x_T, y)$ and OOD distribution as $\tilde{P}(x_I, x_T, y)$ where $\tilde{P}$ may differ from $P$ in terms of the following assumptions.*

- *Case 1: The image and text match, but labels are out of the set, namely:*

$$\mathbb{E}_{P(x_I, x_T)} \left[ \log s(x_I, x_T) \right] = \mathbb{E}_{\tilde{P}(x_I, x_T)} \left[ \log s(x_I, x_T) \right],$$

For every pair $x_I$, $x_T$ and any $\alpha$,

$$\mathbb{E}_{P(y|x_I,x_T)}\left[\log(\alpha s_I(x_I,y)+(1-\alpha)s_T(x_T,y))\right] > \mathbb{E}_{\tilde{P}(y|x_I)}\left[\log(\alpha s_I(x_I,y)+(1-\alpha)s_T(x_T,y))\right],$$

*which means that the ID pairs $x_I$ and $x_T$ should have stronger expressity about the ID label $y$ than OOD pairs.*

- **Case 2:** *The image and text do not match, which we assume:*

$$\mathbb{E}_{P(x_I,x_T)}\left[\log s(x_I,x_T)\right] > \mathbb{E}_{\tilde{P}(x_I,x_T)}\left[\log s(x_I,x_T)\right],$$

*which means the ID pairs should have higher similarity than OOD pairs in this case. For every pair $x_I$, $x_T$ and any $\alpha$,*

$$\mathbb{E}_{P(y|x_I,x_T)}\left[\log(\alpha s_I(x_I,y)+(1-\alpha)s_T(x_T,y))\right] = \mathbb{E}_{\tilde{P}(y|x_I,x_T)}\left[\log(\alpha s_I(x_I,y)+(1-\alpha)s_T(x_T,y))\right],$$

*which means that some ID pairs $x_I$ and $x_T$ may have the same expressity about the label $y$ compared with the OOD pairs.*

- **Case 3:** *The image or text does not match with the labels, which we assume:*

$$\mathbb{E}_{P(y|x_I,x_T)}\left[\log(\alpha s_I(x_I,y)+(1-\alpha)s_T(x_T,y))\right] > \mathbb{E}_{\tilde{P}(y|x_I,x_T)}\left[\log(\alpha s_I(x_I,y)+(1-\alpha)s_T(x_T,y))\right].$$

**Theorem 1** *With Assumption 1, we can show that the proposed DIEAF score satisfies the following:*

$$\mathbb{E}_{\tilde{P}(x_I,x_T,y)}[\log g(x_I,x_T,y)] < \mathbb{E}_{P(x_I,x_T,y)}[\log g(x_I,x_T,y)].$$

**Proof 1** *It is easy to write that:*

$$\mathbb{E}[\log g(x_I,x_T,y)] = \gamma\mathbb{E}_{x_I,x_T}[\log s(x_I,x_T)]+\mathbb{E}_{x_I,x_T}\mathbb{E}_{y|x_I,x_T}[\log([\alpha s_I(x_I,y)+(1-\alpha)s_T(x_T,y)])].$$

*The proof simply follows the assumptions we made for each case. Note that this score only works for positive scores, but sometimes, we may encounter negative scores, and the log may be ill-posed. As a surrogate score function, we eliminate the log and maintain $g(x_I,x_T,y)$ for the same intuition as the above theorem.*

