# OpenReview forum: "'No' Matters: Out-of-Distribution Detection in Multimodality Long Dialogue"
_ICLR.cc/2025/Conference — Submitted to ICLR 2025_

### Official Review · Reviewer_AqfS · 2024-11-03

**Soundness:** 2
**Presentation:** 3
**Contribution:** 1
**Rating:** 3
**Confidence:** 2

**Summary:**

This paper studies the Out-of-Distribution (OOD) detection problem in a multimodal context involving an image and a corresponding dialogue discussing that image. The authors introduce a new scoring framework, based on a visual-language model, to detect mismatches between the dialogue and image input pairs, as well as input pairs with previously unseen labels. The experimental results indicate that this approach outperforms using either modality independently for detection. However, I question the significance of this task. Given the capabilities of current large vision-language models, which are quite powerful, would they not be able to handle any domain images effectively? Why is this task still relevant? Regarding the proposed approach, I found it lacking in novelty and differentiation compared to existing methods. Additionally, the comparisons are outdated, with the most recent being from 2018, which diminishes the credibility of the claims.

**Strengths:**

1. The paper is well-written and clearly presented.
2. Although the proposed scoring function is relatively simple and lacks novelty, the authors provide a detailed analysis and explanation of the intuition behind it.
3. The experiments are extensive and thorough.

**Weaknesses:**

1. The motivation behind the task is unclear, particularly considering the current capabilities of powerful vision-language models, which may already handle OOD scenarios effectively. The relevance of this task remains questionable.
2. The proposed method lacks substantial novelty and does not significantly differentiate itself from prior approaches.
3. The comparisons with existing methods are outdated, with the most recent being from 2018, which weakens the evaluation of the paper's contributions in the context of current research. Additionally, there is no comparison with current large vision-language models (LVLMs) for detection. How do models like GPT-4o, Claude-3.5-Sonnet, Gemini, and Qwen2-VL perform on this classification task?

**Questions:**

1. Why is this task still essential in 2024?
2. How do large vision-language models (LVLMs) perform in direct OOD detection?

---

> ### Author Response · Authors · 2024-11-13
> **Response to Reviewer AqfS**
>
> **Weaknesses**:
>
> - The motivation behind the task is unclear, particularly considering the current capabilities of powerful vision-language models, which may already handle OOD scenarios effectively. The relevance of this task remains questionable.
>
> *Responses*: For the multimodal dialogue domain, it still remains a huge challenge: How to effectively know an image has not or has been mentioned from a long dialogue during people's interaction or from human-agent interactions ([3], [4], [5], [6], [7], [8]). Imagine you are talking with another friend, and during your interactions, you two want to search the related images during your talk contents, while the searched pictures are wrong. I hope this real-life case will help you to understand both from the theory aspect and also from the real-life experience perspective.
>
> - The proposed method lacks substantial novelty and does not significantly differentiate itself from prior approaches.
>
> *Responses*: It would be great to use some evidence to support your statement. For example, what prior approaches in long dialogue OOD detection, and what are these exactly?
>
> - The comparisons with existing methods are outdated, with the most recent being from 2018, which weakens the evaluation of the paper's contributions in the context of current research. Additionally, there is no comparison with current large vision-language models (LVLMs) for detection. How do models like GPT-4o, Claude-3.5-Sonnet, Gemini, and Qwen2-VL perform on this classification task?
>
> *Responses*: Indeed, most of the recent Large VLMs focused on QA tasks. In fact, our framework works for single-round QA (see Results Section in the paper), but our focus on multi-round dialogues from the fact that longer dialogues provide more contextual information in real-life scenarios (e.g., imagine you are talking to someone, it is unlikely that we only have one round of the dialogue, but people will continue talking and interacting [7]), which is crucial in real-life scenarios compared to single-round QA, which typically offers limited contextual information (Singh et al., 2019). Multi-turn dialogues are essential for detecting unseen labels when users engage in conversations related to a picture requested in real-scenario settings. Based on this suggestion, we further tested the proposed score function on single-round QA by sampling the first round from the whole dialogue. The results shown in the Results section indicate that single-round QA carries less information in dialogues. The Gap is: How to improve the performance in more real-life cases without much computing resources cost (We only use one Nvidia 3080 card to run everything).
>
> **Questions**:
> - Why is this task still essential in 2024?
>
> *Responses*: See response to Weakness Point 1.
>
> - How do large vision-language models (LVLMs) perform in direct OOD detection?
>
> *Responses*: Refer to Responses for Weakness Point 3.
>
> **References**:
> - [1] Gan, M., Dou, X., & Jiang, R. (2013). From ontology to semantic similarity: calculation of ontology‐based semantic similarity. The Scientific World Journal, 2013(1), 793091.
> - [2] Saha, A., Khapra, M., & Sankaranarayanan, K. (2018). Towards Building Large Scale Multimodal Domain-Aware Conversation Systems. Proceedings of the AAAI Conference on Artificial Intelligence, 32(1). https://doi.org/10.1609/aaai.v32i1.11331
> - [3] Yang, D., Rao, J., Chen, K., Guo, X., Zhang, Y., Yang, J., & Zhang, Y. (2024, July). IM-RAG: Multi-Round Retrieval-Augmented Generation Through Learning Inner Monologues. In Proceedings of the 47th International ACM SIGIR Conference on Research and Development in Information Retrieval (pp. 730-740).
> - [4] Singh, A., Natarajan, V., Shah, M., Jiang, Y., Chen, X., Batra, D., ... & Rohrbach, M. (2019). Towards vqa models that can read. In Proceedings of the IEEE/CVF conference on computer vision and pattern recognition (pp. 8317-8326).
> - [5] Lang, H., Zheng, Y., Hui, B., Huang, F., and Li, Y. (2023). Out-of-domain intent detection considering multi-turn dialogue contexts. arXiv preprint arXiv:2305.03237.
> - [6] Azizi, S., Culp, L., Freyberg, J., Mustafa, B., Baur, S., Kornblith, S., Chen, T., Tomasev, N., Mitrovi´c, J., Strachan, P., et al. (2023). Robust and data-efficient generalization of self-supervised machine learning for diagnostic imaging. Nature Biomedical Engineering, 7(6):756–779.
> - [7] Gao, R., Roever, C., and Lau, J. H. (2024a). Interaction matters: An evaluation framework for interactive dialogue assessment on English second language conversations. arXiv preprint arXiv:2407.06479.
> - [8] Sima, C., Renz, K., Chitta, K., Chen, L., Zhang, H., Xie, C., Beißwenger, J., Luo, P., Geiger, A. and Li, H., 2023. Drivelm: Driving with graph visual question answering. arXiv preprint arXiv:2312.14150.

---

> > ### Author Response · Authors · 2024-11-26
> > **Fellow-up repsonses to Reviewer AqfS**
> >
> > Dear Reviewer  AqfS,
> >
> > Could you respond based on our previous responses to your concerns?

---

> > > ### Comment · Reviewer_AqfS · 2024-11-28
> > > **Follow-up questions**
> > >
> > > Thank you for your response. Let’s first talk about the main concern I have: is this problem really a challenge for current LVLMs? Is it possible for you to show the results of any popular LVLMs? Proprietary models like GPT-4o, Claude-3.5-Somnet, Gemini-1.5-Pro, or even small open sourced model Qwen2-VL-2B? It would not require too much resources.

---

> > > > ### Author Response · Authors · 2024-11-28
> > > > **Responses to Reviewer AqfS: The focus of this study**
> > > >
> > > > Thank you for raising this important concern about whether our dialogue-image OOD task challenges current LVLMs, which is an important contribution of our study. We need to state that the focus of this study **is not a benchmarking study or evaluation study**, but to improve the OOD detection via a new score function based on the proposed more authentic real-life scenario: multimodal long complex dialogue. Our study uses Clip and Blip. Both models struggled significantly with this OOD scenario, highlighting the difficulty of the task, despite which specific models, as we presented extensive experiments in the results section.
> > > >
> > > > However, it is important to note that Qwen2-VL, or like other LVLMs, still face challenges in generalizing to OOD tasks, especially in complex dialogue-image scenarios with much noise in two modalities **without additional fine-tuning or complex prompting with domain knowledge** [1, 2, 3] , as previous studies have extensively proved these limitations of LVLMs. So, most LVLMs still require substantial fine-tuning to improve their ability to align dialogues with visual data, which our method does not need such complex prompting or combining domain knowledge for fine-tuning or pre-training.
> > > >
> > > > Thus, our contribution lies in defining **a novel more real-life based dialogue-image OOD task**, providing a meaningful **score function** to guide future LVLM research, as the other reviewers recognized.
> > > >
> > > > References:
> > > >
> > > > - [1] Jing, Liqiang, and Xinya Du. "FGAIF: Aligning Large Vision-Language Models with Fine-grained AI Feedback." arXiv preprint arXiv:2404.05046 (2024).
> > > >
> > > > - [2] Zhao, Y., Yin, Y., Li, L., Lin, M., Huang, V.S., Chen, S., Chen, W., Yin, B., Zhou, Z., & Zhang, W. (2024). Beyond Sight: Towards Cognitive Alignment in LVLM via Enriched Visual Knowledge.
> > > >
> > > > - [3] Lin, B., Ye, Y., Zhu, B., Cui, J., Ning, M., Jin, P., & Yuan, L. (2023). Video-llava: Learning united visual representation by alignment before projection. arXiv preprint arXiv:2311.10122.

---

> > > > > ### Comment · Reviewer_AqfS · 2024-12-01
> > > > > **Follow-up response to the authors**
> > > > >
> > > > > Thank you for the response. The three papers reveal the misalignment between text and image modalities and hallucination. However, this is a more basic and general challenge for LVLMs. Your task here is a binary classification to predict whether an image is in-distribution or out-of-distribution regarding the conversation, which is more limited in scope and also less challenging from my point of view. So I believe it is essential to at least test some LVLM's performance on your task to see if this task is really a challenge for LVLMs. As mentioned, perhaps you could test at least a small open-source model like Qwen2-VL-2B?

---

> ### Author Response · Authors · 2024-12-02
> **Follow-up response to Reviewer AqfS: Challenge in the dialogue doamin**
>
> Thanks for your suggestion of using the small open-sourced model to test the performance of our proposed score function. However, as you agreed, the related studies do face a challenge of mismatch between text and visual information, and differentiate from our aim, which is our target is in this study: **improve the match between long conversation and visual pictures**. This is harder and more challenging than the three cited studies (the challenges have been pointed by [1], [2], [3] in future direction as well). However, the performance for Qwen2-7B [1] did not target on **long dialogue matching with visual information**. Once again, to reclaim the current progress in the dialogue domain.
>
> Reference:
> - [1] An Yang, Baosong Yang, Binyuan Hui, Bo Zheng, Bowen Yu, Chang Zhou, Chengpeng Li,
> Chengyuan Li, Dayiheng Liu, Fei Huang, Guanting Dong, Haoran Wei, Huan Lin, Jialong
> Tang, Jialin Wang, Jian Yang, Jianhong Tu, Jianwei Zhang, Jianxin Ma, Jianxin Yang, Jin Xu,
> Jingren Zhou, Jinze Bai, Jinzheng He, Junyang Lin, Kai Dang, Keming Lu, Keqin Chen, Kexin
> Yang, Mei Li, Mingfeng Xue, Na Ni, Pei Zhang, Peng Wang, Ru Peng, Rui Men, Ruize Gao,
> Runji Lin, Shijie Wang, Shuai Bai, Sinan Tan, Tianhang Zhu, Tianhao Li, Tianyu Liu, Wenbin
> Ge, Xiaodong Deng, Xiaohuan Zhou, Xingzhang Ren, Xinyu Zhang, Xipin Wei, Xuancheng
> Ren, Xuejing Liu, Yang Fan, Yang Yao, Yichang Zhang, Yu Wan, Yunfei Chu, Yuqiong Liu,
> Zeyu Cui, Zhenru Zhang, Zhifang Guo, and Zhihao Fan. Qwen2 technical report, 2024.
> https://arxiv.org/abs/2407.10671.
>
> - [2] Ye, G., Zhao, H., Zhang, Z., Zha, X., & Jiang, Z. (2024, June). LSTDial: Enhancing Dialogue Generation via Long-and Short-Term Measurement Feedback. In Proceedings of the 2024 Conference of the North American Chapter of the Association for Computational Linguistics: Human Language Technologies (Volume 1: Long Papers) (pp. 5857-5871).
>
> - [3] Ataallah, K., Shen, X., Abdelrahman, E., Sleiman, E., Zhu, D., Ding, J., & Elhoseiny, M. (2024). Minigpt4-video: Advancing multimodal llms for video understanding with interleaved visual-textual tokens. arXiv preprint arXiv:2404.03413.

---

### Official Review · Reviewer_a5DV · 2024-11-03

**Soundness:** 2
**Presentation:** 3
**Contribution:** 3
**Rating:** 5
**Confidence:** 2

**Summary:**

This paper addresses the challenge of Out-of-Distribution detection in multimodal long-dialogue systems, where text and image modalities are combined, especially in real-world or open-domain dialogue applications. The authors propose the Dialogue Image Aligning and Enhancing Framework, designed to detect two main types(Mismatch between dialogue and image, Unseen labels) of OOD cases.

**Strengths:**

1. This paperintroduces a unique approach for OOD detection by combining image and dialogue data, addressing the limitations of using single modalities.

2. The framework's use of an alignment and enhancement scoring mechanism allows for precise multimodal OOD detection.

3. By focusing on mismatched pairs and unseen labels, the framework is suited for real-world applications where dialogue and visual information often co-occur.

**Weaknesses:**

This paper has chosen multimodal, multi-turn dialogue environments as the task to perform and verify OOD detection. The authors claim this is necessary for user satisfaction and trust, but this part of the argument is not convincing. They need to provide logical supplementation on why multimodal OOD detection tasks are important in dialogue, and why they are particularly important in multi-turn rather than single-turn interactions.
Similar problem definitions were found in other papers [1,2], but references to these papers are missing, and explanations and quantitative metrics are needed to show how this paper differentiates itself from these works. The baseline methods used in this paper are all methodologies from before 2019, and experiments should be designed to include methodologies from recent papers.


[1]. GENERAL-PURPOSE MULTI-MODAL OOD DETECTION FRAMEWORK, V Duong et al. 2023
[2]. MultiOOD: Scaling Out-of-Distribution Detection for Multiple Modalities, H DOng et al. 2024

**Questions:**

1. It would be better to identify and supplement experiments with use cases where multimodal OOD is important and can be well utilized, rather than focusing on the multi-turn dialogue setting.
2. It would be good to add comparative experimental results with recent papers that have proposed solutions to multimodal OOD problems.

---

> ### Author Response · Authors · 2024-11-13
> **Responses to Reviewer a5DV**
>
> Thanks for your review:
>
> **Weaknesses Responses**: We would appreciate these two papers recommended by the reviewer. However, after carefully reading the two works, the definitions of these two works are quite different compared with our work, our focus is how to improve the OOD label detection for multi-modality long open domain conversation with images, which is more real-life focused. And in the past two years, no single studies have focused on multimodality multi-round long conversations in OOD detection works, even in the NLP dialogue domain ([5], [6], [7]). Most works are related in single-round VQA ([8]), so it is impossible for us to find baselines in past two years. Actually for some datasets, we checked the quality ourselves by manually checking (e.g., ImageChat, and DialogCC, and PhotoChat), this process took more than two-month by the authors (we can provide all the dataset checking evidence if required).
>
> **Questions**:
> - It would be better to identify and supplement experiments with use cases where multimodal OOD is important and can be well utilized, rather than focusing on the multi-turn dialogue setting.
>
> *Responses*: Indeed, our framework definitely works for single-round QA (see Results in the paper), but our focus on multi-round dialogues from the fact that longer dialogues provide more contextual information in real-life scenarios (e.g., imagine you are talking to someone, it is unlikely that we only have one round of the dialogue, but people will continue talking and interacting [7]), which is crucial in real-life scenarios compared to single-round QA, which typically offers limited contextual information (Singh et al., 2019). Multi-turn dialogues are essential for detecting unseen labels when users engage in conversations related to a picture requested in real-scenario settings. Based on this suggestion, we further tested the proposed score function on single-round QA by sampling the first round from the whole dialogue. The results shown in the Resluts section indicate that single-round QA carries less information in dialogues.
>
> - It would be good to add comparative experimental results with recent papers that have proposed solutions to multimodal OOD problems.
>
> *Responses*: In the past two years, no single studies have focused on multimodality multi-round long conversations in OOD detection works, even in the NLP dialogue domain ([5], [6], [7]). Most works are related in single-round VQA ([8]).
>
> **References**:
> - [1] Gan, M., Dou, X., & Jiang, R. (2013). From ontology to semantic similarity: calculation of ontology‐based semantic similarity. The Scientific World Journal, 2013(1), 793091.
> - [2] Saha, A., Khapra, M., & Sankaranarayanan, K. (2018). Towards Building Large Scale Multimodal Domain-Aware Conversation Systems. Proceedings of the AAAI Conference on Artificial Intelligence, 32(1). https://doi.org/10.1609/aaai.v32i1.11331
> - [3] Yang, D., Rao, J., Chen, K., Guo, X., Zhang, Y., Yang, J., & Zhang, Y. (2024, July). IM-RAG: Multi-Round Retrieval-Augmented Generation Through Learning Inner Monologues. In Proceedings of the 47th International ACM SIGIR Conference on Research and Development in Information Retrieval (pp. 730-740).
> - [4] Singh, A., Natarajan, V., Shah, M., Jiang, Y., Chen, X., Batra, D., ... & Rohrbach, M. (2019). Towards vqa models that can read. In Proceedings of the IEEE/CVF conference on computer vision and pattern recognition (pp. 8317-8326).
> - [5] Lang, H., Zheng, Y., Hui, B., Huang, F., and Li, Y. (2023). Out-of-domain intent detection considering multi-turn dialogue contexts. arXiv preprint arXiv:2305.03237.
> - [6] Azizi, S., Culp, L., Freyberg, J., Mustafa, B., Baur, S., Kornblith, S., Chen, T., Tomasev, N., Mitrovi´c, J., Strachan, P., et al. (2023). Robust and data-efficient generalization of self-supervised machine learning for diagnostic imaging. Nature Biomedical Engineering, 7(6):756–779.
> - [7] Gao, R., Roever, C., and Lau, J. H. (2024a). Interaction matters: An evaluation framework for interactive dialogue assessment on English second language conversations. arXiv preprint arXiv:2407.06479.
> - [8] Sima, C., Renz, K., Chitta, K., Chen, L., Zhang, H., Xie, C., Beißwenger, J., Luo, P., Geiger, A. and Li, H., 2023. Drivelm: Driving with graph visual question answering. arXiv preprint arXiv:2312.14150.

---

> > ### Comment · Reviewer_a5DV · 2024-11-22
> > **response to author**
> >
> > Thank you for your reply
> > . While I appreciate that this paper focuses on multi-turn OOD detection and introduces the novel topic of multi-turn multimodal dialogue OOD detection, I don't think the methodology fundamentally leverages the inherent differences between multi-turn and single-turn dialogues. The methods presented could equally be applied to single-turn scenarios (which the authors consider similar to VQA) for OOD detection, rather than specifically addressing the unique characteristics of multi-turn interactions.
> >
> >
> > For instance, there are numerous distinctive aspects of dialogue that differ between single-turn and multi-turn interactions. One clear example is the increasing ambiguity that emerges as the number of turns grows - this phenomenon manifests quite differently in single-turn versus multi-turn contexts.
> >
> > However, I remain very positive about the paper's problem formulation and would recommend several improvements:
> > 1. Further justification for using multi-turn dialogue as the primary testbench
> > 2. Greater emphasis on exploring the various distinctions between single-turn and multi-turn dialogues
> > 3. Development of methods that specifically address these unique multi-turn characteristics
> > These enhancements would strengthen the paper's contribution to the field of multi-turn dialogue OOD detection.

---

> ### Author Response · Authors · 2024-11-22
> **Second response to Reviewer a5DV**
>
> Thank you for your comments on our paper. We sincerely appreciate your positive recognition of our problem formulation regarding solving this gap and your suggestions.
>
> - Addressing Methodology Concerns:
> As we addressed in previous responses, single-turn QA only provides limited information for the dialogue compared with multi-turn QA, which is more common in social user interactions [1], so the major purpose of this study is to address how to dealing with multi-round dialogues (we choose Real MMD and Visdial dataset for experiments). While our methodology can indeed be adapted to single-turn scenarios, we designed it with multi-turn dialogues in mind to explore the complex dynamics unique to multi-turn interactions to align with more real-world scenarios, and we do show that our method can handle it with the tables given in the results section (we also attached the table here for your convenience) . From the attached table, it can be seen that signle round QA are limited in matching with visual information, and more rare in real-life (e.g., chatting to an online assistant for shopping and finding the right pictures of the products in long dialogues ). The increasing ambiguity as the number of turns grows is an important phenomenon, and we acknowledge that further elaboration on how our methods address this would strengthen the contribution.
>
> ### Table 1: Single Round (SR) QA Comparison (FPR95/AUROC/AUPR)
> |Score|Agg|Dialogue|Dialogue SR|Ours|Ours SR|
> |-|-|-|-|-|-|
> |MSP|Max|76.9/66.5/48.8|89.0/65.4/49.2|73.4/73.2/53.5|84.4/69.2/56.1|
> |Prob|Max|67.9/73.5/56.1|87.1/67.3/51.1|55.3/78.8/57.9|74.8/74.1/60.9|
> | |Sum|91.9/62.3/45.7|94.7/62.2/45.4|72.8/73.6/56.6|78.4/71.4/56.4|
> |Logits|Max|67.9/73.5/56.1|87.1/67.3/51.1|57.2/82.6/72.7|62.6/78.9/68.1|
> | |Sum|98.6/44.1/36.0|98.4/50.1/40.0|97.2/49.9/37.4|97.2/53.1/40.8|
> |Odin|Max|72.1/73.2/55.5|89.6/67.0/50.7|59.6/78.9/58.8|76.6/73.6/61.6|
> | |Sum|91.9/61.6/45.2|94.9/61.5/45.0|73.0/73.2/56.0|79.7/70.5/55.4|
> |Mahalanobis|Max|66.0/75.8/56.8|71.7/63.9/42.0|49.7/83.2/67.1|60.3/81.8/66.7|
> | |Sum|78.6/68.6/50.0|81.2/60.8/40.4|75.0/76.2/60.2|77.0/74.3/57.6|
> |Energy|Max|67.9/73.5/56.1|82.4/68.8/52.0|57.6/82.5/72.6|60.6/79.5/68.2|
> | |Sum|67.0/74.1/57.1|87.1/67.3/51.1|55.9/82.3/72.2|63.4/78.8/68.1|
> |Average|Max|69.8/72.7/54.9|84.5/66.6/49.4|58.8/79.9/63.8|69.9/76.2/63.6|
> | |Sum|85.6/62.1/46.8|91.3/60.4/44.4|74.8/71.0/56.5|79.1/69.6/55.7|
>
>
> -  Emphasis on Multi-Turn-Specific Applied Methods: We will improve our narrative by enhancing our methodology with enhancements that directly leverage the unique characteristics of multi-turn interactions, ensuring that these methods cannot be trivially applied to single-turn contexts. Further justification for Multi-Turn Focus: We will provide a deeper rationale for selecting multi-turn dialogues as the primary benchmark, illustrating how this choice allows us to address real-world challenges more effectively.
>
> References:
> - Moon, S., He, H., Jia, H., Liu, H., & Fan, J. W. (2023). Extractive Clinical Question-Answering With Multianswer and Multifocus Questions: Data Set Development and Evaluation Study. JMIR AI, 2(1), e41818.

---

> > ### Author Response · Authors · 2024-11-26
> > **Fellow-up responses**
> >
> > Dear Reviewer a5DV,
> >
> > Could you respond based on our previous responses for your concerns?

---

> > > ### Author Response · Authors · 2024-11-28
> > > **Fellow-up responses: seeking for active engagement**
> > >
> > > Could you respond based on our previous responses for your concerns? Thank you.

---

> > > > ### Comment · Reviewer_a5DV · 2024-12-03
> > > >
> > > > Thank you for your response. I am confident that if the authors supplement these points, they can establish the legitimacy of selecting multi-turn dialogue as their primary verification environment. However, such changes are not minor, which need substantial changes, and I will maintain my current score of 5 points for this draft state.

---

### Official Review · Reviewer_ziNZ · 2024-11-04

**Soundness:** 2
**Presentation:** 3
**Contribution:** 2
**Rating:** 5
**Confidence:** 3

**Summary:**

This paper presents a new framework, Dialogue Image Aligning and Enhancing Framework (DIAEF), to improve the user experience in multi-round dialogues by efficiently detecting out-of-distribution (OOD) instances in multimodal contexts, specifically dialogue-image pairs. DIAEF integrates visual language models with novel scoring mechanisms to identify OOD cases in two main scenarios: mismatches between dialogue and image inputs and previously unseen labels in input pairs. Experimental results show that the combined use of dialogue and image data enhances OOD detection more effectively than using each modality independently, demonstrating robustness in prolonged dialogues. This approach supports adaptive conversational agents and sets a benchmark for future research in domain-aware dialogue systems.

**Strengths:**

- This paper is well-written, especially considering that the topic of OOD detection is not easy to understand. For instance, the authors explain the problem formulation of "cross-modal OOD detection" clearly.
- The paper introduces a new paradigm and framework for OOD detection in "multi-turn interactive dialogue," along with a new scoring method, DIAEF, which utilizes vision-language models.
- They demonstrate the effectiveness of DIAEF both experimentally and theoretically and suggest its potential as an alternative scoring method, as shown in Table 1.
- Through extensive experiments, the authors show that DIAEF outperforms other OOD detection scoring methods and empirically validate their design choices (e.g., the selection of alpha).

**Weaknesses:**

- Although the authors clearly present the problem formulation of "cross-modal OOD detection," I still find the use of OOD terminology in the multi-modal dialogue domain unclear. Dialogue inherently has a subjective nature and a one-to-many structure (i.e., diversity [1]), meaning that even with the same query, there are multiple possible responses depending on the situation and the user in real-world interactions. Therefore, I question whether using the term "OOD" is appropriate in this context. The authors should further clarify why handling OOD detection in the multi-modal domain is necessary.
- Additionally, I am concerned that using CLIP or BLIP models may not ensure adequate understanding of dialogue, as CLIP has a limited context length of 77 tokens, and neither CLIP nor BLIP is pretrained on open-domain dialogue datasets—issues highlighted in prior works [2-3]. When determining OOD, it seems that the embedding model reflects its training distribution, yet CLIP embeddings may be ineffective for dialogue. I believe that using LongCLIP [4] could be a better alternative. Therefore, the authors should clarify their choice of CLIP or BLIP for the VLM models.
- In the DIAEF framework, training the "label extractor" is crucial; however, I don’t fully understand what constitutes a "label" in an "open-domain dialogue." Could you explain this?
- While the authors demonstrate the effectiveness of their framework, more experiments are needed to establish its robustness and reliability across additional dialogue datasets. The framework formulation includes multiple hyperparameters (e.g., $\alpha$ and $\gamma$), and the MMD dataset is not a high-quality multi-modal dialogue dataset since it is synthesized using CLIP matching, despite the application of human crowdsourcing to verify contextual relevance. Nevertheless, this dataset lacks both high quality and diversity, which is mentioned in the prior work [5]. I recommend that the authors conduct experiments on additional dialogue datasets, such as PhotoChat [6], MP-Chat [7], ImageChat [8], and DialogCC [5]. Given time constraints, it is unnecessary to experiment on the full datasets; subsampled versions would suffice.
- I am also curious as to why the authors focus on "long dialogue," as, to my knowledge, the datasets used in the experiments emphasize single-session dialogues rather than multi-session dialogues like MSC [9] or Conversational Chronicles [10].

---

**References**

[1] Li, Jiwei, et al. "A diversity-promoting objective function for neural conversation models." arXiv preprint arXiv:1510.03055 (2015).

[2] Yin, Zhichao, et al. "DialCLIP: Empowering Clip As Multi-Modal Dialog Retriever." ICASSP 2024-2024 IEEE International Conference on Acoustics, Speech and Signal Processing (ICASSP). IEEE, 2024.

[3] Lee, Young-Jun, et al. "Large Language Models can Share Images, Too!." arXiv preprint arXiv:2310.14804 (2023).

[4] Zhang, Beichen, et al. "Long-clip: Unlocking the long-text capability of clip." arXiv preprint arXiv:2403.15378 (2024).

[5] Lee, Young-Jun, et al. "Dialogcc: Large-scale multi-modal dialogue dataset." arXiv preprint arXiv:2212.04119 (2022).

[6] Zang, Xiaoxue, et al. "Photochat: A human-human dialogue dataset with photo sharing behavior for joint image-text modeling." arXiv preprint arXiv:2108.01453 (2021).

[7] Ahn, Jaewoo, et al. "Mpchat: Towards multimodal persona-grounded conversation." arXiv preprint arXiv:2305.17388 (2023).

[8] Shuster, Kurt, et al. "Image chat: Engaging grounded conversations." arXiv preprint arXiv:1811.00945 (2018).

[9] Xu, J. "Beyond goldfish memory: Long-term open-domain conversation." arXiv preprint arXiv:2107.07567 (2021).

[10] Jang, Jihyoung, Minseong Boo, and Hyounghun Kim. "Conversation chronicles: Towards diverse temporal and relational dynamics in multi-session conversations." arXiv preprint arXiv:2310.13420 (2023).

**Questions:**

Please refer to Weaknesses.

---

> ### Author Response · Authors · 2024-11-13
> **Response to Reviewer ziNZ**
>
> Thank you for bringing your perspectives:
>
> - Definition of OOD: Refer to the introduction on Page 1. We presented the rationale and definitions based on previous studies in the dialogue domain.
>
> - The purpose of this study are not about the image caption, but try to improve the robustness of OOD detection between long-dialogues and images, **despite the embeddings of specific models**, otherwise, it's just a model-specific related solution without robustness to more general VLM models.
>
> - Based on linguistics theory and NLP dialogue studies supported by Gan and Jiang (2013), the design of our label extractor is rooted in theories emphasizing the importance of both semantic meanings (contextual meaning of a label's token in dialogue) and ontological distances (how humans understand and process certain tokens in languages). These two aspects are considered equally important in our label selection process. For instance, the Label 'Cat' can refer to an animal, but in a different context, such as a dialogue about singers, it could refer to 'Doja Cat'. The real meaning of a label in real-life dialogues heavily depends on context. WordNet evaluates the similarity between different meanings of 'Cat' based on various semantic relations. This approach helps the OOD label by considering the relationships between words and their meanings. Indeed, we explored other tuning strategies and learning weights for the selection presented in Table 2.
>
> - It would be great to click on the link at the bottom of the Second Page: https://anonymous.4open.science/r/multimodal_ood-E443/README.md , with the detailed information of the datasets and all the information in our dialogue datasets, including a multi-round QA, and long real-life open-domain conversations. Our selection of the datasets is based on previous works ([2],[3],[4]) and the most suitable cases for dialogue-image detection ([5], [6]). Actually for some reviewer's proposed datasets, we also checked the quality ourselves by manually checking the details (e.g., ImageChat, and DialogCC, and PhotoChat), this process took *more than two-month* by the authors in this study (we can provide all the dataset checking evidence if required).
>
> -  Indeed, our framework definitely works for single-round QA (see Results in the paper), but our focus on multi-round dialogues from the fact that longer dialogues provide more contextual information in real-life scenarios (e.g., imagine you are talking to someone, it is unlikely that we only have one round of the dialogue, but people will continue talking and interacting [7]), which is crucial in real-life scenarios compared to single-round QA, which typically offers limited contextual information (Singh et al., 2019). Multi-turn dialogues are essential for detecting unseen labels when users engage in conversations related to a picture requested in real-scenario settings. Based on this suggestion, we further tested the proposed score function on single-round QA by sampling the first round from the whole dialogue. The results shown in Resluts section indicate that **single-round QA carries less information in dialogues**.
>
> **References:**
> - [1] Gan, M., Dou, X., \& Jiang, R. (2013). From ontology to semantic similarity: calculation of ontology‐based semantic similarity. The Scientific World Journal, 2013(1), 793091.
> - [2] Saha, A., Khapra, M., & Sankaranarayanan, K. (2018). Towards Building Large Scale Multimodal Domain-Aware Conversation Systems. Proceedings of the AAAI Conference on Artificial Intelligence, 32(1). https://doi.org/10.1609/aaai.v32i1.11331
> - [3] Yang, D., Rao, J., Chen, K., Guo, X., Zhang, Y., Yang, J., & Zhang, Y. (2024, July). IM-RAG: Multi-Round Retrieval-Augmented Generation Through Learning Inner Monologues. In Proceedings of the 47th International ACM SIGIR Conference on Research and Development in Information Retrieval (pp. 730-740).
> - [4] Singh, A., Natarajan, V., Shah, M., Jiang, Y., Chen, X., Batra, D., ... & Rohrbach, M. (2019). Towards vqa models that can read. In Proceedings of the IEEE/CVF conference on computer vision and pattern recognition (pp. 8317-8326).
> - [5] Lang, H., Zheng, Y., Hui, B., Huang, F., and Li, Y. (2023). Out-of-domain intent detection considering
> multi-turn dialogue contexts. arXiv preprint arXiv:2305.03237.
> - [6] Azizi, S., Culp, L., Freyberg, J., Mustafa, B., Baur, S., Kornblith, S., Chen, T., Tomasev, N., Mitrovi´c,
> J., Strachan, P., et al. (2023). Robust and data-efficient generalization of self-supervised machine
> learning for diagnostic imaging. Nature Biomedical Engineering, 7(6):756–779.
> - [7] Gao, R., Roever, C., and Lau, J. H. (2024a). Interaction matters: An evaluation framework
> for interactive dialogue assessment on English second language conversations. arXiv preprint
> arXiv:2407.06479.

---

> > ### Comment · Reviewer_ziNZ · 2024-11-26
> > **Response by Reviewer**
> >
> > Thanks for addressing my questions. By the way, the link provided by the authors is expired. Can you share it once again?

---

> > > ### Author Response · Authors · 2024-11-26
> > > **Fellow-up responses for solving the concerns**
> > >
> > > Here's the attached link: https://anonymous.4open.science/r/multimodal_ood-E443/README.md
> > >
> > > Thanks for your acknowledgment of solving your concern.

---

> > > > ### Comment · Reviewer_ziNZ · 2024-11-28
> > > > **Response by Reviewer**
> > > >
> > > > Thank you for sharing the link. Based on the authors' responses, I now have a clearer understanding of their work. I will raise my original score from 3 to 5.

---

### Official Review · Reviewer_RMqc · 2024-11-04

**Soundness:** 2
**Presentation:** 3
**Contribution:** 2
**Rating:** 5
**Confidence:** 4

**Summary:**

This paper take the first attempt for OOD detection in multimodality long dialogue, propose a framework that enhances the OOD detection in cross-modal contexts，achieved the combination of OOD detection and multimodal methods.

And it demonstrate that integrating image and multi-round dialogue OOD detection is more effective with previously unseen labels than using either modality independently.

**Strengths:**

The starting point chosen for the paper is quite innovative.

**Weaknesses:**

The innovation of the methods used in the paper needs to be strengthened.

**Questions:**

none

---

> ### Author Response · Authors · 2024-11-13
> **Response to Reviewer RMqc**
>
> **Weaknesses**:
> - The innovation of the methods used in the paper needs to be strengthened
>
> Responses: It is unreasonable and even outrageous to make such a statement of 'lack of novelty' and not give any specific feedback or rationals. This is a far lower than a qualified standard review.

---

### Official Review · Reviewer_UswD · 2024-11-04

**Soundness:** 2
**Presentation:** 2
**Contribution:** 2
**Rating:** 5
**Confidence:** 3

**Summary:**

This paper addresses the challenge of out-of-distribution (OOD) detection in multimodal contexts, particularly focusing on the combined input of dialogues and images in real-life applications such as open-domain conversational agents. It introduces the Dialogue Image Aligning and Enhancing Framework (DIAEF), an approach for detecting mismatches in dialogue and image pairs and identifying previously unseen input labels in conversations. DIAEF integrates visual language models with scoring metrics tailored for two primary OOD scenarios: (1) detecting mismatches between dialogue and image inputs, and (2) flagging dialogues with previously unseen labels. Experiments conducted on several benchmarks indicate that DIAEF’s integrated approach to image and multi-round dialogue OOD detection outperforms single-modality methods, especially in dialogues involving mismatched pairs and extended conversations.

**Strengths:**

- The paper focuses on two key types of out-of-distribution (OOD) scenarios: (1) mismatches between dialogue and image inputs, and (2) inputs with previously unseen labels. It demonstrates the effectiveness of the proposed method in accurately identifying these OOD cases.
- This work marks the first attempt to address OOD detection in dialogue contexts, specifically for multi-round conversations. To support this, the authors constructed a new dataset for multi-round question-answering, enabling comprehensive evaluation of the framework’s performance in real-life dialogue settings.

**Weaknesses:**

- Models like CLIP and BLIP are primarily trained for image captioning, and some previous researches suggest that they may not generate optimal text embeddings for dialogue. How does this paper address the potential limitations of using these models in a dialogue context to ensure accurate and meaningful embeddings?
- Does the proposed method consider only yes/no question-answer dialogues as in-domain scenarios? If so, when OOD situations become more complex, it’s unclear how well the method would perform or if it would remain effective in identifying out-of-domain cases accurately.

**Questions:**

Can you include the example of the test set generated?

---

> ### Author Response · Authors · 2024-11-13
> **Responses to Reviewer UswD**
>
> **Weaknesses**:
> - Models like CLIP and BLIP are primarily trained for image captioning, and some previous researches suggest that they may not generate optimal text embeddings for dialogue. How does this paper address the potential limitations of using these models in a dialogue context to ensure accurate and meaningful embeddings?
>
> Responses: The purpose and focus of this study are not about caption, but to try to improve the OOD detection ability between dialogues and imagines. This miss-understand the whole focus of this study.
>
> - Does the proposed method consider only yes/no question-answer dialogues as in-domain scenarios? If so, when OOD situations become more complex, it’s unclear how well the method would perform or if it would remain effective in identifying out-of-domain cases accurately.
>
> Responses: This is a classical dialogue-related issue. If you take a closer look at the dataset we included in the paper, the dialogue is long and complex in most cases instead of just YES/NO short-format questions.
>
> **Question**:
>
> - If you click on the link at the bottom of the second page, which is: https://anonymous.4open.science/r/multimodal_ood-E443/README.md . You will see all the detailed information of the dataset, and with all the information in our dialogue datasets.

---

> > ### Author Response · Authors · 2024-11-26
> > **Fellow-up responses for Reviewer UswD**
> >
> > Dear Reviewer,
> >
> > Could you respond to our responses to your concerns?

---

> > > ### Author Response · Authors · 2024-11-28
> > > **Seeking responses for Reviewer UswD**
> > >
> > > Dear Reviewer,
> > >
> > > Could you respond to our responses to your concerns?

---

### Author Response · Authors · 2024-11-13
**Responses to all review**

We felt sorry to see the qualification of this year's review based on the following reasons:

- Reviewers (e.g., In this review, Reviewer RMqc gave a 3-score with only providing **one 10-word sentence** reviewing feedback without any specific feedbacks, Reviewer AqfS gave a 3-score with challenging the OOD task in 2024), all demonstrated a lack of understanding or even the reading of this paper, and the answers/solutions to most of these feedback questions can be easily found in the abstract and introduction sections (the first two pages).

- The reviews demonstrated a lack of understanding of the dialogue domain and led to the huge issue of domain knowledge understanding in the significance of this study (Reviewer AqfS gave a 3-score on this study, while with a limited understanding for prior approaches in multimodal dialogue domain for OOD detection task， and didn't provide enough reasons why this proposed method is the same compared with prior approaches). For the dialogue domain, tackling this question is a huge challenge, and that's why we employed such extensive experiments to ensure the performance.

- Based on previous reasons, the reviewing marks do not align with feedback for Reviewer AqfS and Reviewer RMqc. For example, these two reviewers give 3 without pointing out a strong rationale. We sincerely hope to see a rational justification with actual reasons on the score and we would love to improve with enough justifications based on scientific review.

---

### Meta-Review · Area_Chair_Qkko · 2024-12-18

**Metareview:**

This paper proposes a framework for Out-of-Distribution (OOD) detection in multimodal, multi-turn dialogue scenarios, integrating image and dialogue inputs with a novel scoring method. While the approach is clearly explained and supported by experiments, the reviewers found issues that limit the impact. Some reviewers questioned the fundamental motivation and relevance of OOD detection in this setting, given the capabilities of current advanced vision-language models. Others (a5DV, AqfS) criticized the lack of comparison with recent methods and datasets, undermining claims of novelty. Although the paper attempts a new angle on multimodal OOD detection, the limited clarity on the task's current relevance, insufficient novelty, outdated comparisons, and uncertain applicability to genuine dialogue scenarios constrain its contribution. I recommend rejection.

**Additional Comments On Reviewer Discussion:**

During the discussion period, the authors expressed disappointment in the reviewers responses.  Low-quality reviews are considered carefully in the decision process (UswD, RMqc).
I did not highly consider feedback from reviewers who did not engage during this phase. Among those who did comment back, the critiques focused on the questionable relevance of the OOD task in the era of powerful LVLMs, a lack of comparisons with more recent methods, and the unclear significance of applying OOD detection to multi-turn dialogues. Reviewers who engaged still found the paper's task motivation weak, its benchmarks outdated, and its methodological distinctions unclear. Thus, even after considering the rebuttal, the core issues remain unresolved.

---

### Decision · Program_Chairs · 2025-01-22

Reject